# Effects of Heat-Moisture Treatment on the Digestibility and Physicochemical Properties of Waxy and Normal Potato Starches

**DOI:** 10.3390/foods12010068

**Published:** 2022-12-23

**Authors:** Guihong Fang, Ke Liu, Qunyu Gao

**Affiliations:** 1Carbohydrate Laboratory, School of Food Science and Engineering, South China University of Technology, Guangzhou 510640, China; 2Heinz Mehlhorn Academician Workstation, Department of Nutrition and Food Hygiene, International School of Public Health and One Health, Hainan Medical University, Haikou 571199, China

**Keywords:** waxy potato starch, normal potato starch, heat-moisture treatment, physicochemical properties, digestibility

## Abstract

Heat-moisture treatment (HMT) is a safe, environmentally friendly starch modification method that reduces the digestibility of starch and changes its physicochemical properties while maintaining its granular state. Normal potato starch (NPS) and waxy potato starch (WPS) were subjected to HMT at different temperatures. Due to erosion by high-temperature water vapor, both starches developed indentations and cracks after HMT. Changes were not evident in the amylose content since the interaction between the starch molecules affected the complexation of amylose and iodine. HMT increased pasting temperature of NPS from 64.37 °C to 91.25 °C and WPS from 68.06 °C to 74.44 °C. The peak viscosity of NPS decreased from 504 BU to 105 BU and WPS decreased from 384 BU to 334 BU. The crystallinity of NPS decreased from 33.0% to 24.6% and WPS decreased from 35.4% to 29.5%. While the enthalpy values of the NPS declined from 15.74 (J/g) to 6.75 (J/g) and WPS declined from 14.68 (J/g) to 8.31 (J/g) at 120 °C. The solubility and swelling power of NPS decreased while that of WPS increased at 95 °C. Due to the lack of amylose in WPS, at the same HMT processing temperature, the reduction in peak viscosity of treated WPS compared to that of native starch was smaller than that of NPS. The resistant starch (RS) content of NPS after HMT at 120 °C was 73.0%. The slowly digestible starch (SDS) content of WPS after HMT at 110 °C was 37.6%.

## 1. Introduction

According to the International Diabetic Federation (IDF), 537 million people are currently living with diabetes, and this number is projected to increase to 783 million by 2045 [1]. Type 2 diabetes (T2D) has become a significant health concern worldwide. Indigestible carbohydrates, such as resistant starch (RS), are associated with a lower glycemic index (GI) and a reduced risk of T2D [2]. The potential benefits of RS include improved glucose tolerance, higher insulin sensitivity, and increased post-meal satiety. Moreover, RS may benefit health since it is an important substrate for short-chain fatty acid synthesis [3]. Recently, RS has attracted considerable attention as a healthy dietary fiber and can be obtained with physical, chemical, enzymatic, and genetic methods. Different methods can produce RS with different properties [4]. Different preparation processes can rearrange the amylose and amylopectin in starch, modifying its morphology, rheological properties, gelatinization parameters, X-ray patterns, crystallinity double helical structure, swelling force, and water binding ability [5].

The HMT is a common physical method to improve the RS content of starch. It involves exposing starch granules to high temperatures (80–120 °C) at low moisture levels (10–30%) for a specific period of time between 15 min and 16 h [6]. Compared to modern pregelatinization techniques, HMT consumes far less energy, is easy to use, and does not require expensive equipment [7]. Compared to chemical techniques, HMT is an environmentally friendly, efficient modification method [8]. Various studies have utilized HMT to treat maize [9], rice [10,11], cassava [12], wheat [7,13], potato [14], and elephant foot yam starch [15]. Suriya et al. [15] found that the granules exhibited granular aggregation and fissures and cavities on the granule surfaces, and exhibited higher gelatinization temperatures and lower gelatinization enthalpy than the elephant foot yam starch granules after HMT. Trung et al. [14] indicate that the formation of RS was significantly different among the sweet potato starches in which the yellow sweet potato starch had lower RS content after HMT. After HMT treatment, the properties of different varieties of starch will change differently. Yet, researchers have paid little attention to waxy starch. Normal and waxy starch display different levels of change in starch structure and properties after HMT due to variations in the amylose content and amylopectin chain mobility [16]. Jiranuntakul [17] et al. studied three normal and three waxy varieties of starch subjected to prolonged exposure to low temperatures. The results showed a more pronounced HMT effect on the normal starch than the waxy starch after HMT at 100 °C for 16 h. They did not address the change law of the varieties of starches with short exposures at different temperatures. HMT modification is not only affected by internal factors, such as plant sources and amylose content, but also by external processing conditions, such as processing temperature, processing time, and moisture content. There has never been a comparative analysis of the changes in two different starches under the same preparation conditions at different temperatures. This study compared normal potato starch (NPS) and waxy potato starch (WPS) after HMT at different temperatures while analyzing the digestive property and internal starch molecule changes. In this paper, the effect of different temperatures on the change law of phytochemicals and digestibility of potato from different varieties were investigated.

## 2. Materials and Methods

### 2.1. Materials 

The NPS was donated by Huaou Starch Co., Ltd., Huhehaote, China, while the WPS was purchased from AVEBE, Veendam, The Netherlands. The porcine trypsin (P7545) and amyloglucosidase (A7095) were purchased from Sigma-Aldrich Chemical Co. (St. Louis, MI, USA), while the glucose oxidase-peroxidase (GOPOD) assay kit was obtained from Megazyme International Ireland Ltd. (Wicklow, Ireland). All other chemical reagents were of analytical grade.

### 2.2. Starch Modification with HMT

Starch (20 g) was weighed, and water was added until the starch moisture content reached 25%. The sample was sealed in a stainless-steel reactor, equilibrated at room temperature for 24 h, and allowed to react for 2.5 h at 100 °C, 110 °C, and 120 °C, respectively. The samples were dried in an oven at 45 °C and sieved using a 100-mesh sieve. The NPS samples treated with HMT at 100 °C, 110 °C, and 120 °C were labeled NPS100, NPS110, and NPS120, while the corresponding WPS samples were labeled WPS100, WPS110, and WPS120, respectively. Native normal potato starch (NNPS) and native waxy potato starch (NWPS) represented the two native starches. 

### 2.3. Scanning Electron Microscopy (SEM) 

A scanning electron microscope (EVO18, Zeiss, Oberkochen, Germany), operated at an accelerating voltage of 10–20 kV, was used to observe the morphology of the starch granules. The starch was evenly dispersed on conductive adhesive and sprayed with gold for 3 min (Balzers Med 010 Buenos Aires, Argentina), after which its morphology was observed at 500× magnification, and a suitable viewing angle was selected [18].

### 2.4. Polarized Light Microscopy

A small amount of 1% starch suspension was collected and deposited onto glass slides. The focal length of the polarizing microscope was adjusted, and a suitable viewing angle was selected at 200× natural light, which was then adjusted to polarized light conditions to observe the Maltese crosses of different starches [18].

### 2.5. Solubility and Swelling Power

The swelling degree was measured using a method similar to the one described by Yassaroh [19]. Starch was converted into a 2% starch slurry, which was dispersed evenly and allowed to react in a constant temperature water bath (65 °C, 80 °C, and 95 °C) for 30 min while shaking every 5 min. The mixture was cooled to room temperature and centrifuged for 15 min. The supernatant was decanted into petri dishes and dried at 105 °C. The dry matter and precipitate were weighed after centrifugation, and the solubility (100%) and swelling power (*g*/*g*) were calculated as follows:(1)$$\mathrm{Solubility}\;\left(100\%\right)=\frac{\mathrm{mass}\;\mathrm{of}\;\mathrm{dried}\;\mathrm{mater}*100}{\mathrm{mass}\;\mathrm{of}\;\mathrm{dry}\;\mathrm{starch}\;}\;$$
(2)$$\mathrm{Swelling}\;\mathrm{power}\;\left(g/g\right)=\frac{\mathrm{precipitate}\;\mathrm{weight}}{\mathrm{mass}\;\mathrm{of}\;\mathrm{dry}\;\mathrm{starch}*\;\left(1-\mathrm{solubility}\right)}\;$$

### 2.6. Amylose Content

The iodine absorbance method was used to determine the amylose content according to the national standard GB/T15683-2008 determination of amylose content in rice. The 100 mg starch sample (dry basis) was mixed with 95% ethyl alcohol (1 mL) and 1 M sodium hydroxide (9 mL). To induce gelatinization, the mixture was heated and cooled to room temperature. Next, the starch solution was transferred to a 100 mL volumetric flask and diluted with distilled water to 100 mL. From that, 5 mL were taken, and volume was adjusted to 50 mL with diluent. Subsequently, 1 mL acetic acid and 2 mL of iodide (KI) and iodine (I_2_) were added, and then diluted with distilled water to 100 mL. The solution was allowed to stand for 10 min at ambient temperature prior to absorbance measurements at 620 nm. The amylose content was calculated from a standard curve plotted for mixtures of amylose and amylopectin from potatoes (0–100% amylose).

### 2.7. Differential Scanning Calorimetry (DSC)

Water was added to 3 mg dry basis starch until reaching a total weight of 10 mg. The gold plate was sealed and equilibrated at room temperature for more than 24 h. The temperature was maintained at 30 °C for 1 min, after which it was increased to 120°C at a rate of 10 °C/min and then cooled from 120 °C to 30 °C at the same rate. All samples were measured at least in duplicate. The onset temperature (T_o_), peak temperature (T_p_), and conclusion temperature (T_c_) were recorded. The enthalpy of gelatinization (ΔH) was estimated by integrating the area between the thermogram and the subpeak baseline, which was expressed in joules per gram (J/g) of dry starch [8]. 

### 2.8. Pasting Properties

The paste viscosity of the samples was measured using a Microvisco-Amylograph (Brabender, Bremen, Germany). A 6% starch suspension was heated from 30 °C to 95 °C, which was maintained for 5 min, and cooled to 50 °C, which was maintained for 5 min. The heating and cooling rates were 7.5 °C/min.

### 2.9. X-ray Diffraction 

The prepared samples were measured using an X-ray diffractometer (D8 Advance, Bruker, Steermark, Austria) at 40 kV and 40 mA with Cu Kα radiation (λ = 0.154 nm). Sample powder was packed tightly in rectangular glass cells and scanned at an angle range of 5° to 40° [7].

### 2.10. In-Vitro Digestion

The starch samples were digested refer to a method described by Englyst et al. [20]. The enzyme solutions were prepared by dispersing pancreatin in sodium acetate buffer at pH 5.2, followed by centrifugation for 10 min at 4500 rpm after mixing well. The supernatant was mixed with 0.67 mL of amyloglucosidase to obtain the mixed enzyme for later use. Approximately 300 mg of the sample and 5 glass balls were equilibrated for 10 min at 37 °C and stirred (170 r/min) with 15 mL sodium acetate buffer (pH 5.2). The samples were then treated with a 0.75 mL pancreatin and amyloglucosidase mixture. Next, 0.5 mL aliquots of the hydrolyzed solution were collected after 20 min and 120 min, respectively. Then, 10 mL of anhydrous ethanol was added to each aliquot to deactivate the enzymes. The glucose content was determined after centrifugation (4500 rpm for 5 min) using the GOPOD assay kit. The rapidly digestible starch (RDS), SDS, and RS were calculated by combining the glucose released at 20 min and that released at 120 min.

### 2.11. Statistical Analysis

The Origin Program 8.0 (Origin Laboratory Company, Northampton, MA, USA) was used to analyze and report the mean values and standard deviations. An analysis of variance (ANOVA) was conducted, followed by Duncan’s multiple-range test using SPSS 19.0. The significance level was set at *p* < 0.05.

## 3. Results and Discussion

### 3.1. Morphology of the Starch Granules

SEM allows for a convenient and intuitional characterization of starch granules. Various studies have shown that no obvious grain starch deformation is evident after HMT modification [21,22]. The SEM images of the native starch and its HMT sample are shown in Figure 1, while those of the polarized light microscopy are shown in Figure 2. The SEM images indicated that the granule surfaces of the two native starches were smooth with typical Maltese cross patterns, as shown in Figure 2. Except for cracks and indentations on the surfaces of some of the particles, the granular structures were intact with no obvious gelatinization after HMT. A similar phenomenon was observed by Wang [23,24]. As shown in Figure 2, the cross-refraction inside some particles disappeared at a higher temperature. The faded central birefringence indicated a loss of radial orientation due to the increased mobility of the double helices during HMT [25]. HMT can transfer or rearrange the molecular structure at the center of the starch granules where the tissue structure is weak. This phenomenon occurred mainly because water molecules entered the interior of the particles to form hydrogen bonds with the molecular chain of the starch during HMT, which was confirmed by the polarizing microscope image. Starch with higher amylopectin content is more sensitive to HMT was confirmed. The subsequent physicochemical characterization results were also consistent with result. Suriya [15] found that the cross-section of each starch granule displayed a large hollow region at the center, possibly caused by HMT. Similar observations have been reported for bean starch [26], rice [24], and buckwheat [27] starch exposed to HMT, while oat starch [28] retains a smooth surface, which may be related to different species and HMT conditions.

### 3.2. Swelling Power

The swelling power can reflect the bonding degree of the internal bonds in the particles to some extent. The starch expanded from a relatively loose amorphous region, followed by expansion in an amorphous region near the crystalline region, and finally, the expansion of the crystalline region [4]. The swelling power of NPS and WPS before and after HMT treatment is shown in Figure 3. The swelling power of NNPS increased with the increase in temperature, with the highest amount at 95 °C. The swelling power reflects the water absorption capacity of starch granules during gelatinization [29]. The swelling power of the NPS decreased after HMT at three different temperatures. This phenomenon also occurred in oat [28], corn [30], and buckwheat starches [31]. HMT conditions may reinforce the internal interactions between molecular chains, hindering the hydration of free hydroxyl groups [32], while a more compact side chain cluster contributes to a rigid crystalline architecture that inhibits water percolation into the starch granules. Additionally, HMT-induced amylose complexes can reduce amylose leaching through their ability to prevent starch swelling [11]. The swelling power of NWPS was far higher than the modified starch at 65 °C but lower at 80 °C and 95 °C. HMT increased the WPS gelatinization temperature. At 95 °C, the swelling power of WPS100, WPS110, and WPS120 was significantly higher than that of the NWPS. After HMT, the gelatinization temperature of WPS was lower than 80 °C. The swelling power increased significantly when the water bath temperature reached 80 °C and 95 °C, causing granule swelling and gelatinization. This may be because the crystal and molecular structures of the starch are damaged to varying degrees during HMT, allowing water molecules to more readily enter the starch granules at high temperature. Therefore, starch with a different amylose content of the same variety presents different hydration capacities.

### 3.3. Solubility

The solubility of the NPS and WPS starch was reduced after HMT at 65 °C and 80 °C. The solubility of WPS increased significantly after HMT at 95 °C compared to NPS (Figure 4). The reduction in the solubility of the HMT starch could be attributed to additional interactions between the amylose-amylopectin (AM-AP) and amylose-amylose (AM-AM) chains during HMT, reducing the dissolution of amylose [33]. Suriya et al. [14] and Li et al. [32] both reported that HMT reduced the starch solubility below 80 °C. At 95 °C, WPS after HMT was completely gelatinized, and the double helical structure was partially untwisted, the structure became loose, and more water-soluble short chains were dissolved.

### 3.4. Amylose Content

Although HMT may increase the amylose fragments [28], our study shows that the amylose content was not affected by HMT (Table 1). It is possible that this is due to a short time of HMT, which has less influence on amylose dissolution. HMT decreased the content of amylose in starch granules, and the effects were significantly proportional to the duration of HMT [34]. Amylose content did not change significantly after 2 h of treatment. Varatharajan et al. [16] believed that the decrease in amylose content was primarily ascribed to the Am-Am and Am-AP interactions, reducing the ability of amylose to form a single helix. A single helix and iodine form a blue complex concentration in response to amylose content. Studies have also found decreased amylose content in other types of starch, such as cereals. Manuel [35] and Hyun [36] believed that this decline was due to the formation of starch-lipid complexes during HMT. Harinder et al. [37] also revealed a decrease in the amylose content of samples after HMT. However, the effect of conditions in this study on the amylose content of NPS and WPS was not obvious.

### 3.5. Thermal Properties 

The energy (endothermic) absorbed during the gelatinization was monitored with DSC. The enthalpy characteristics of the two types of starches varied significantly after HMT, while the T_o_, T_p_, and T_c_ increased (Table 2), which was consistent with the results obtained by Na [38]. The endothermic changes were attributed to a new surface layer formed during HMT, restricting water penetration into the granules and delaying swelling [39]. A reduction in ΔH and an increase in T_o_, T_p_, and T_c_ indicated that starch granule gelatinization became difficult after HMT. Li [26] believed this was because HMT partially changed the starch crystal structure, which dissolved, destroyed, and promoted the interaction between molecular chains, resulting in new crystal formation. It was highly likely that the B crystals melted to form A-type crystallites during the polymorphic transformation, this was confirmed in the following XRD. NPS crystal formation occurred due to the interaction and recombination of Am–AP and amylopectin–amylopectin (AP-AP) chains, while only AP–AP chain interaction and recombination was necessary for WPS crystal formation. T_c_–T_o_ indicated the degree of difference within the starch granules, such as the structure of the double helix and the size of the microcrystalline crystals. The enthalpy of the starch decreased significantly after HMT, mainly due to the degradation of the double helical structure, while water promoted the interaction between the molecular chains, limiting the mobility of the amorphous region [19]. The enthalpy of NPS and WPS after HMT was significantly lower than NNPS and NWPS, indicating that the dissociation and melting ability of amylopectin were hindered by amylose-amylopectin and amylopectin–amylopectin interaction in NPS and amylopectin–amylopectin interaction in WPS [14]. Lan [40] proved that the ΔH decrease in the well-ordered portion after HMT modification was below 10.3%, while that in the less-ordered portion reached 85.8%~100%.

### 3.6. Pasting Properties

The viscograph was used to determine the pasting properties of starches. Temperature significantly affects the paste properties. The pasting temperature of the two processed starches increased compared to the native samples (Figure 5). As a result, HMT either strengthened the interaction between starch molecules or increased the thermostability of starch. The property changes after HMT were ascribed to the association between the chains of the amorphous region of the granules and the changes in crystallinity during the thermal process [41]. After HMT at different temperatures, the pasting temperature gradually increased at a higher treatment temperature, indicating that the starch granules were more resistant to swelling [32]. The peak viscosity, breakdown, and setback values were lower than the native starch, while the heat stability and starch shearing improved significantly. The HMT temperature was positively correlated with this decline [42]. In the same HMT conditions, the changes in the viscosity characteristics of WPS were smaller than that of NPS. This was mainly due to a lack of amylose in the waxy granules. WPS have less chance of forming a stable, dense double helix than NPS during HMT [43].

### 3.7. XRD Pattern

XRD confirmed that the starch crystallinity could be divided into A, B, and C types, and was affected by amylopectin content, average chain length, the orientation of the double helices to the X-ray beam, and crystallite size [44]. The X-ray diffraction patterns of the different samples and their corresponding relative crystallinity are shown in Figure 6, indicating that the two native samples contained B crystals. B-type X-ray profiles are commonly found in root starch, amylose-rich starch, cereal tubers, and some fruit seeds with peak intensities of 5.5°, 14.4°, 17.2°, 22.2°, and 24° [45]. The two native starches exhibited a peak value at 20°, indicating that the starch formed a V-type inclusion complex with lipids. At a diffraction angle of 5.5°, the strength of NNPS exceeded that of NWPS, while the same phenomenon was evident at 22°–24°. This may be due to the presence of type A unit cells in NWPS [16]. After HMT, the crystals of the two starches were transformed from B-type to A-type, while the relative crystallinity decreased gradually. The results were consistent with Tan [46]. NPS100 retained the weak characteristic B crystal peak. A weak diffraction intensity was apparent at 5.5°, with two less obvious peaks at 23°, while no peak was evident at 15°. As the reaction temperature increased, the partial characteristic B crystal peaks of NPS110 and NPS120 gradually disappeared. However, the characteristic type A intensity peaks at 15° and 23° were higher and more visible for the HMT starches, especially for NPS120. The peak at 5.5° of WPS100 disappeared, while that at 15° was more obvious. Kim [47] believed that the decrease in peak intensity at 5.5° was due to crystal reorganization rather than the destruction of the double helical structure forming the crystal array. The intensity peak of WPS at 15° increased gradually as the reaction temperature increased. These phenomena indicate that WPS is more susceptible to HMT, corresponding with the results presented in Figure 1. Furthermore, WPS was more prone to pits and voids in the center of the polarized cross. The crystalline form of the two starches changed from B to A after HMT, indicating that the structure of the B crystalline form was unstable, with low heat tolerance and poor thermal stability. HMT transformed this into a more stable crystal form [48]. 

### 3.8. The Impact of HMT on Starch Digestibility

Starch digestibility is influenced by two factors. One is the ability of amylolytic enzymes to bind to starch granules, and the other factor concerns how many binding sites are present and how they are surrounded [49]. As shown in Figure 7, the RDS content in the NNPS was lower than in NWPS, while the RS level was higher, which may be related to the difference in amylose content in the two native starches. The amylose content is negatively correlated with the digestibility of starch [50]. After treatment, the RDS content in the two starches decreased, which was consistent with Yang [51]. It is likely that molecular chains adhere to an orderly packing after HMT that makes them less susceptible to enzymatic breakdown, as evidenced by the inhibited digestibility. In addition, the realignment is resistant enough to survive gelatinization. However, the RDS content showed a more significant decline in WPS than NPS, showing that HMT had an obvious impact on starch digestibility, additionally with a more substantial effect on that of WPS. Starch of higher amylopectin-to-amylose ratio is more sensitive to HMT, because steric hindrance near α-(1,6) glycosidic bond is weaker in comparison with that near α-(1,4) bond [52]. After HMT, dense double helix could protect starch glycosidic bonds from enzyme attack. The contact between amylase and starch molecules would become more difficult. which is in line with the previous XRD results. Compared with the native starches, the HMT samples displayed higher RS content, while the SDS varied according to the starch source and HMT temperature. WPS were more likely to form SDS after HMT. A debranched amylopectin chain may realign into an ordered region with structural rigidity, which increases enzyme resistance [46]. There is a possibility that SDS may yield due to interaction between external branch chains A and B1 that possess maximum DP ≥ 15.5 [53]. NPS were more likely to form RS after HMT. As presented in the literature, RS and SDS primarily comprise a (semi-)crystalline region formed by double helices whose DP is 13~24 [32]. Recombination of the starch molecules occurs preferentially in amylose or long chain amylopectin, which could promote RS formation [54]. Furthermore, the RS level increased at a higher temperature, while the SDS content reduced. This shows that starch after HMT under high temperature was more likely to form resistant starch, and after HMT under low temperature was more likely to form SDS. As a result, starch with different digestibility can be intelligently prepared by changing the process conditions or varieties in the future.

## 4. Conclusions

WPS and NPS are compared to better understand how HMT affects their physico-chemical properties. Under the same treatment conditions, WPS were more sensitive than NPS to HMT. Increasing the value of amylopectin to amylose makes starch granules active by absorbing more water during treatment, which promotes their expansion and favors morphological changes driven by thermal force. Compared with the original starch, the swelling degree and solubility of the two treated starches decreased, while the solubility and swelling of WPS increased at 95 °C because of completely gelatinization. After HMT, the gelatinization temperature and thermal stability of the two starches increased while the enthalpy decreased. This means HMT is able to offer a more stable and resistant architecture, allowing it to be used in products that are easy to age. Hot paste stability and swell power of WPS after HMT were superior. Hence, it is used as a thickener to increase the smoothness and stability of caramel and gummy. The crystalline type of the starch is converted from type B to type A, while the crystallinity is decreased after HMT. Furthermore, the higher the amylose content, the more RS formed during HMT and, in contrast, the higher SDS content. The physical and chemical properties of the two starches change after HMT at the same temperatures, indicating that the presence of amylose affects the amylopectin reorganization. The fine structure of amylopectin and amylose is a key factor forming resistant starch. More attention can be paid to the nano-level changes in HMT starch granules and the specific locations of degraded molecular chains. In the future, starch raw materials and processing conditions can be selected according to the demand for starch digestibility to achieve the intelligent control of starch digestibility. Due to the high RS content, HMT-treated potato starch can be used to develop low GI foods for glycemic control in patients with diabetes mellitus. Considering the compositional complexity of food, the changes in the properties of the complexes composed of starch and other components, such as proteins, fats, and polyphenols, after HMT require further investigation. Furthermore, the influence of the HMT heating source, working pressure, and cooling procedure on the properties of starch needs exploration.

## Figures and Tables

**Figure 1 foods-12-00068-f001:**
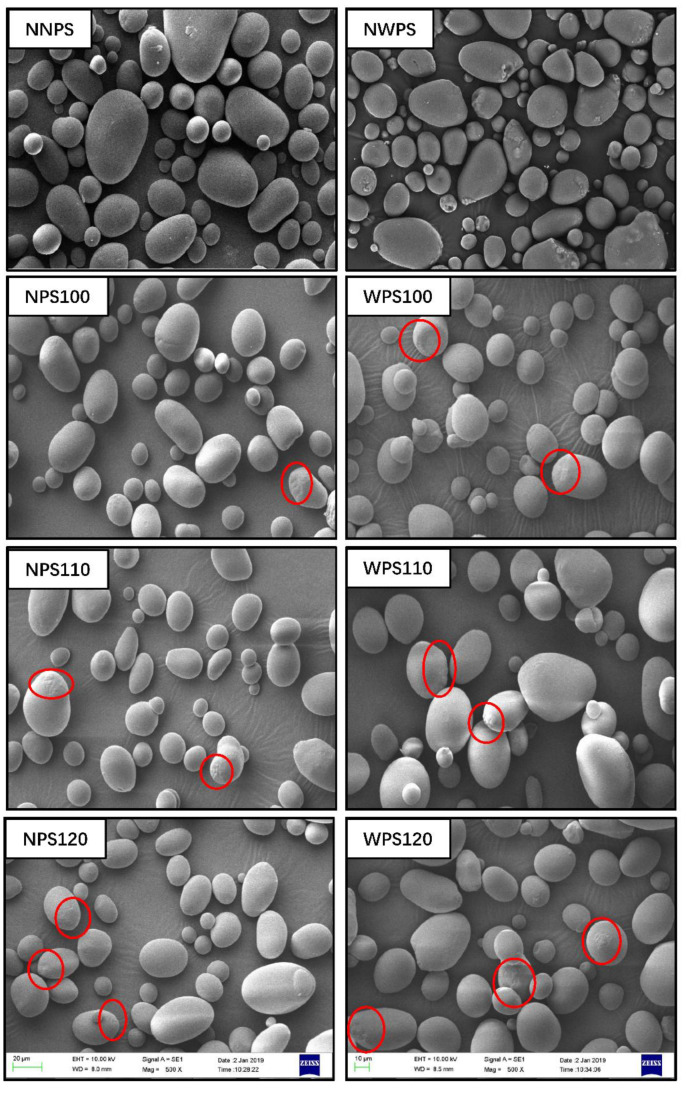
Scanning electron micrographs of the samples (500×) (Red circles were indicated change in the particle surface).

**Figure 2 foods-12-00068-f002:**
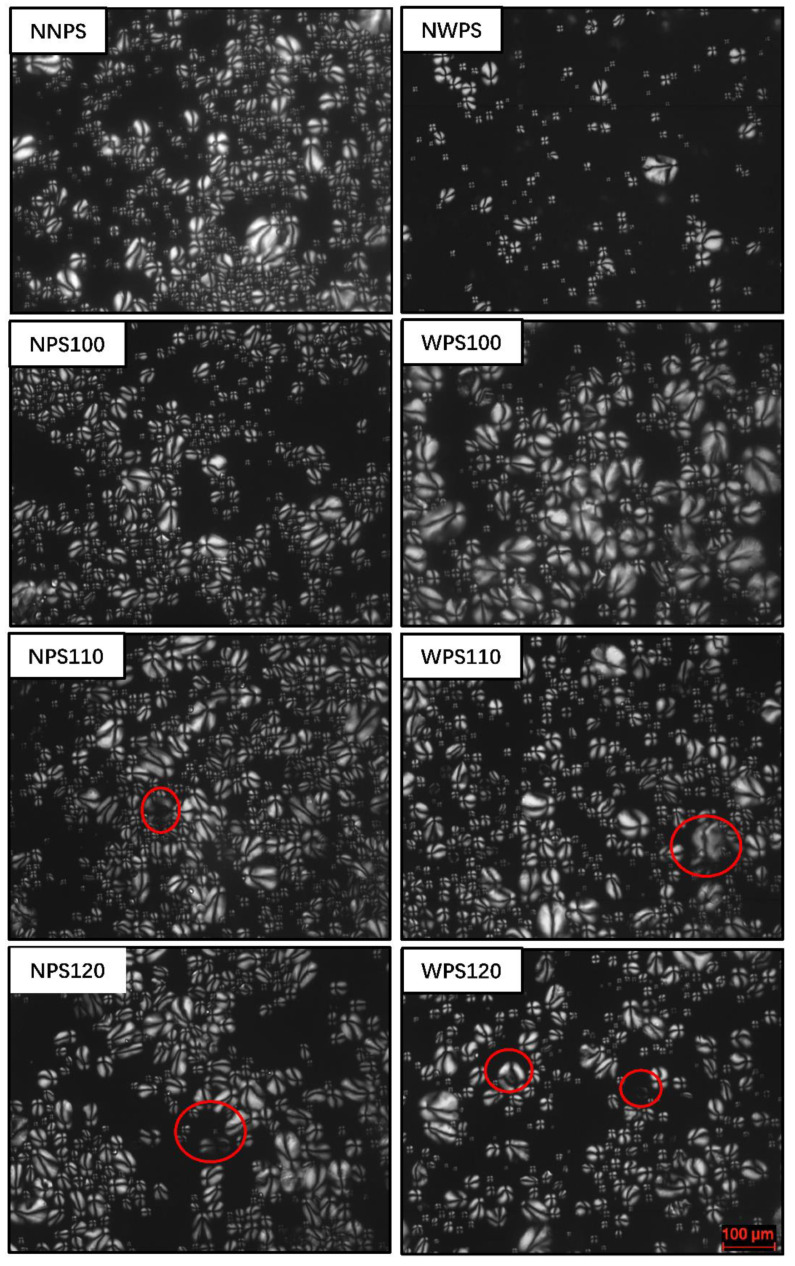
Polarized light microscopy of the samples (200×) (Red circles were indicated change in light refracted by the particles).

**Figure 3 foods-12-00068-f003:**
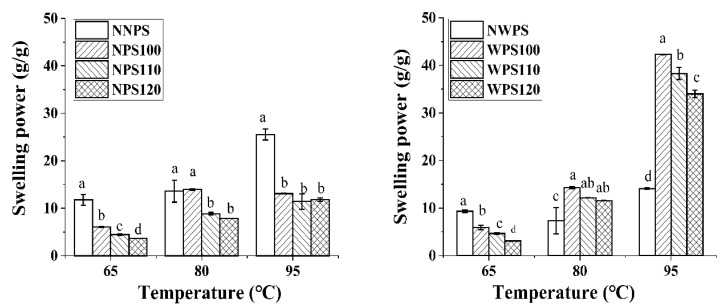
The swelling power of the samples. Different letters in a histogram signify significant differences (*p* < 0.05).

**Figure 4 foods-12-00068-f004:**
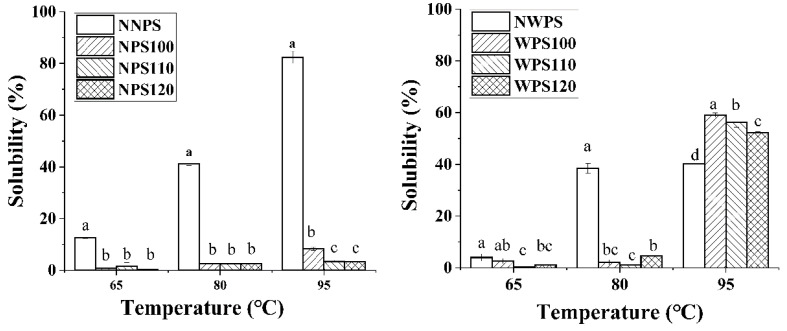
The solubility of the samples. Different letters in a histogram signify significant differences (*p* < 0.05).

**Figure 5 foods-12-00068-f005:**
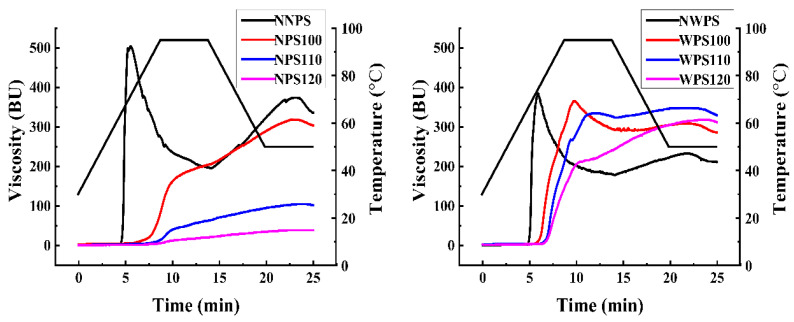
The pasting profiles of the samples.

**Figure 6 foods-12-00068-f006:**
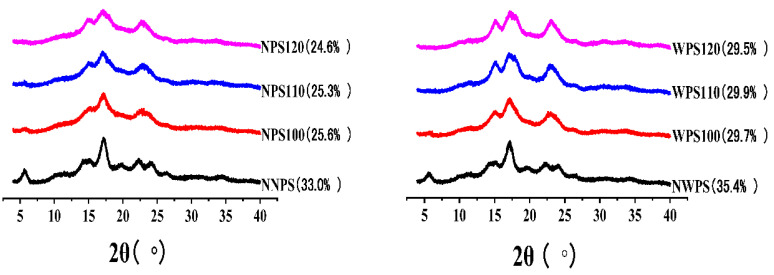
X-ray diffraction patterns of the samples.

**Figure 7 foods-12-00068-f007:**
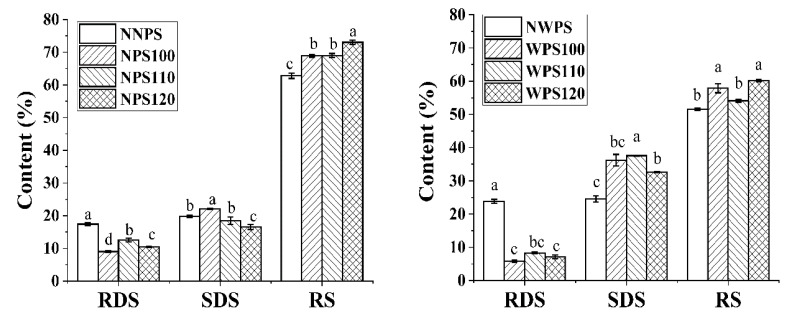
Digestibility of the samples. Different letters in a column signify significant differences (*p* < 0.05).

**Table 1 foods-12-00068-t001:** The amylose content in the samples.

Sample	Amylose Content (%)	Sample	Amylose Content (%)
NNPS	40.68 ± 0.23 ^a^	NWPS	3.40 ± 0.38 ^a^
NPS100	40.63 ± 0.31 ^a^	WPS100	3.02 ± 0.30 ^a^
NPS110	39.49 ± 0.10 ^a^	WPS110	3.02 ± 0.00 ^a^
NPS120	40.68 ± 1.31 ^a^	WPS120	2.70 ± 0.46 ^a^

Different letters in a column denote significant differences (*p* < 0.05). Data were expressed as averages ± standard deviations.

**Table 2 foods-12-00068-t002:** The thermal properties of the samples.

Sample	T_o_ (°C)	T_p_ (°C)	T_c_ (°C)	T_c_–T_o_ (°C)	ΔH (J/g)
NNPS	64.37 ± 1.67 ^d^	68.90 ± 2.09 ^c^	76.57 ± 2.34 ^c^	12.20 ± 0.67 ^a^	15.74 ± 0.67 ^a^
NPS100	76.31 ± 0.21 ^c^	84.43 ± 0.00 ^b^	94.46 ± 0.64 ^b^	18.15 ± 0.86 ^a^	9.29 ± 1.18 ^b^
NPS110	82.16 ± 0.64 ^b^	87.95 ± 0.80 ^ab^	96.12 ± 0.11 ^b^	14.14 ± 0.27 ^a^	7.73 ± 1.87 ^b^
NPS120	91.25 ± 1.26 ^a^	99.67 ± 5.34 ^a^	110.41 ± 5.42 ^a^	19.16 ± 4.16 ^a^	6.75 ± 1.27 ^b^
WPS	68.06 ± 0.08 ^b^	73.27 ± 0.00 ^a^	79.90 ± 0.07 ^b^	11.85 ± 0.15 ^b^	14.68 ± 0.60 ^a^
WPS100	69.61 ± 0.23 ^b^	82.97 ± 0.11 ^a^	88.21 ± 0.08 ^ab^	18.61 ± 0.15 ^a^	8.52 ± 1.53 ^b^
WPS110	71.58 ± 0.80 ^a^	82.03 ± 7.40 ^a^	93.90 ± 1.14 ^a^	22.32 ± 0.34 ^a^	9.40 ± 0.45 ^b^
WPS120	74.44 ± 1.94 ^a^	82.58 ± 1.03 ^a^	92.78 ± 4.65 ^a^	18.34 ± 2.71 ^a^	8.31 ± 2.13 ^b^

Different letters in a column denote significant differences (*p* < 0.05). Data were expressed as averages ± standard deviations.

## Data Availability

The data presented in this study are available on request from the corresponding author.

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
