# Peer review of "Effects of Heat-Moisture Treatment on the Digestibility and Physicochemical Properties of Waxy and Normal Potato Starches"

_foods, 2022, doi:10.3390/foods12010068_

Round 1
Reviewer 1 Report
This manuscript seems to be well prepared, and only a small number of corrections weill be required. The overall quality of language is good, and only a relatively small polishing will be needed.
More detailed comments
l. 67-74 sample code - please use in whole the uniform notation for samples. In some instances it is NNPS (NWPS) in other NPS (WPS). I prefer the second one
l. 91 surface dishes - you mean Petri dishes?
l. 96 Amylose content - can you provide a brief description of applied method. Is it suitable for potato starch
l. 99 DSC please re-write this section. It not clear
l. 128 RDS, SDS, and RS - please describe them when first time used
l. 151 and later on consider the replacement of "may" by "can"
fig. 1 & 2 - to be honest, it is really hard to spot the differences. Also captions has to corrected - you presented micrograph/photos/images etc
l. 165 ...
| The swelling power of NPS and WPS before and after HMT ???? |
l. 181 variety or type or origin
l. 191 Suriya et al. [14} Li et al. [32]
l. 207 There are some problems with references, namely Chung and Hoover [35] 34 is Manuel ...
l. 219 for sure it is not [37]
l. 227 Am-AMP - please describe it when first time used. Also you can use AM and AP oa Am and Amp (Ap)
l. 240 Table 2 - for SP and solubility NPS and WPS starches werestatistically evaluated separately. Here you decided to analyze them both. Such treatment is rather odd
l. 245 The Brabender is a name of companvy. Except MicoVisco... the are manufacturing also E-type and other equipment
l. 279-281 So what about l. 237-9. Can you provide a consistent explanation?
l. 317 The Maltese character - you mean birefringence
Reviewer 2 Report
In the introduction authors should provide some results of HTM, what did the treatment modify in the different studies? Authors simply just mention to which products the treatment has been applied without providing any deeper insight. Only one study has some details. Also, the authors should make clear the reason why there are making this modification, what is the overall aim? What problem will it tackle? Also, I suggest changing the word “normal” potato starch to something else, isn’t “native” more appropriate?
I have concerns about the methodology, isn’t 2.5 hours way too much time? Also, at such high temperatures, didn’t a lot of water evaporate? Was this quantified to know the exact concentration of the solution after treatment? Many methodologies need further details. Also, it is not mentioned how many replicates were carried out for each treatment. Also, the author’s main point is that HTM changes the amylose content, however, this was not found in their study. Authors say it can be due to the method that does not detect amylose that is bound to other compounds. Shouldn’t another method have been tested then?
Results need to be discussed deeper and compared more to the literature and other studies. Especially the discussion on starch digestibility is very poor.
The conclusion needs to be re-written as it is simply a long summary of the results, and this is not what is expected in a conclusion.
Other minor comments:
Abstract: Provide some quantitative results as well
Line 40: define HMT at first use. Before comparing it to other methods I suggest the authors give a quick overview of what is this method
Line 41: and does not require expensive equipment
Line 45: Reformulate. Yet, little….
Line 53: Punctuation
Line 67: Be specific, which mass? How many grams?
Line 82: emulsion? When and how was the emulsion prepared?
Line 88: Dry basis starch? What is this?
Line 98: More details about this methodology should be given.
Line 100: Why was the starch hydrated for DSC analysis? Why not carry out the analysis with the powder?
Line 138: Intuitive? What is meant by that?
Reviewer 3 Report
The manuscript has investigated the effects of heat-moisture treatment on the physicochemical properties and digestibility of native and waxy potato starches. The manuscript is well-written and the results are interesting. However, the manuscript is not acceptable in its present form and should be revised.
Comments:
Line 17: Delete “treatment”
Line 82: starch emulsion?! Do you mean starch suspension?
Line 112: Please explain how you have calculated the relative crystallinity.
Line 161 swelling power: Please explain why the swelling power was increased by increasing the heating temperature in all of the treatments. Moreover, add the papers with similar findings. You can use the following manuscript: 10.3390/gels8110693
Fig 3: The caption should be modified; Please write “temperature” instead of “column”
Line 199-200: The statistical analysis did not show significant differences; it means that there were not any differences (not a minimal decline). Please delete this sentence and write “The amylose content was not affected by HMT”.
Line 244: In the “swelling power” test you have reported that the swelling power of HMT waxy starch was higher than native waxy starch. However, in “pasting properties” the peak viscosity of the control sample is higher than theHMT waxy starch. What is the main reason for this behavior?
Line 261: Please cite other papers who have reported the transformation of crystalline pattern from B to A in HMT starches.
Conclusion: Please revise the conclusion; it is a summary of the results. You should highlight the major findings, their importance, and the potential applications of HMT starches based on their properties.
Round 2
Reviewer 2 Report
The manuscript has been considerably revised, yet there are still some points that need clarification. Also, English, punctuation, and use of capital and lower letter should be revised thoroughly.
In the abstract, I still miss some quantitative results. For instance, instead of “ decrease” use decreased from X to Y, and the same goes for everywhere where it reads “increased”.
The authors should emphasize in the introduction in what their study differs from Jiranuntakul [17] et al.
It should be clear what is the added value of the present study.
2 reviews commented that the conclusion is not appropriate as it is a huge summary. This huge summary remains unchanged. As already mentioned, authors should highlight the major findings, their importance, and the potential applications of HMT starches based on their properties.
Other comments:
Line 18: To vague as “high temperature” is relative. Write at temperatures higher than X
Line 41: Define HTM at first use
Line 45: Repetition: “and does not require expensive equipment, and requires no expensive equipment”
Line 47: remove comma
Line 55: rephase as it is not understandable
Line 98: Capital letter
Line 113: Revise
Line 159: Still awkward, “more convenient and intuitional” as compared to what?
Line 162: Further discuss SEM images as it is only reported and not discussed.
Figure 2: Indicate what are the red circles in the images
Reviewer 3 Report
The manuscript is acceptable.
Author Response
Thank you very much for your time involved in reviewing the manuscript and your very encouraging comments on the merits.